# Cox Proportional Hazards Regression for Interval-Censored Data with an Application to College Entrance and Parental Job Loss

HeeJin Kim [ID], Sunghun Kim [ID] and Eunjee Lee [ID]*

Department of Information and Statistics, Chungnam National University, Daejeon 34134, Korea
* Correspondence: eunjee.cnu@gmail.com, Tel.: +82-42-821-5435

**Abstract:** This study involved conducting a survival analysis by fitting a Cox proportional hazards model to Korea Labor Panel data to analyze the impact of parental job loss on children's delayed admission to colleges and universities in South Korea, using 376 subjects whose parental education levels were college-level or higher. Since Korea Labor Panel data are interval- and right-censored, we compared three imputation methods: simple omission, imputation as the average of the left and right values of the interval, and multiple imputation. Their integrated areas under the ROC curve (AUC) and mean square errors (MSE) were compared to assess their predictive and estimation performances. It was found that, within the simulation, the multiple imputation method exhibited a lower MSE than the other two methods. However, no difference was observed in the iAUC values. In the group where each householder had at least a college degree, parental job loss was significantly related to the delayed college or university admission of the first-born child regardless of the use of the interval censoring imputation method. In particular, when the first-born children experienced their parents' unemployment at the age of 18, the probability of college admission was reduced nearly by 53% compared to cases where they did not. This analysis implies that the government should develop a policy in the education system offering psychosocial support for adolescents who cannot expect parental help.

**Keywords:** parental job loss; college entrance; survival analysis; Cox proportional hazards rate; interval censoring; multiple imputation

## 1. Introduction

Parental job loss affects children's educational achievement. Unemployment results in an immediate income loss and thus obliges parents to reduce financial support for their children's education, thereby negatively affecting their school performance (Becker and Tomes 1986; Berger et al. 2009; Blau 1999; Rege et al. 2011). Job loss can deteriorate a child's family environment through family separation, alcohol-related problems in mothers, and increased mortality in fathers (Mörk et al. 2020). Such disruptions in their family environments hinder children's educational achievements (Codjoe 2007; Muola 2010; Pan and Ost 2014; Parveen 2007).

Educational attainment is one measure of educational achievement (Nielsen and Roos 2015). Since it is critical to social and economic success beyond efforts and ability (Goldthorpe 1996), many researchers have studied the relationships between parental job loss and children's college enrollment (Coelli 2011; Lindemann and Gangl 2019; Pan and Ost 2014). Coelli (2011) investigated the relationship between parental job loss when children are in the age group of 16 to 18 years old and their enrollment in university and community college. University enrollment is a binary outcome for youth: enrolled in a university or community college in the two years after high school. They considered youth enrollment in the two years after graduating high school as an outcome. They found that youth who have experienced parental job loss tend to go to university in later years,

and not immediately after high school, to make the money necessary for the tuition fee. Pan and Ost (2014) analyzed the difference in the college admission rates of adolescents depending on the difference in their parents' unemployment period. All the households in this paper experienced parental unemployment. The experimental group experienced parental unemployment at 15–17 years of age, and the control group experienced parental unemployment at 21–23 years of age. Their analysis confirmed that the experimental group's college entrance rate was 10% lower than that of the control group. Lindemann and Gangl (2019) found that paternal unemployment during children's secondary education had a negative effect on their enrollment in postsecondary education.

We investigated if parental job loss contributes to delayed college enrollment in South Korea. A Cox regression model was employed to analyze how the timing of college enrollment is affected by parental job loss (Cox 1972). Previous studies simply focused on whether the affected children were able to enroll in college or not at the age of 18 or 19 years, which was a binary outcome observed in a short term. Meanwhile, we used the data for the analysis based on 20 years of panel data from surveys conducted between 1998 and 2017 as part of the Korean Labor & Income Panel Study (KLIPS). The timing of college enrollment was determined during the follow-up time, which ranged from one year to ten years and varied with subjects. By doing so, we could utilize information about delayed college enrollment to analyze the effects of parental unemployment. For example, if the outcome variable is college enrollment of a child aged 18 or 19 years, a child who enrolls in college three years after high school graduation is considered "not enrolled." Meanwhile, in our analysis, the child's outcome was three, the exact timing of college enrollment. Through this analysis, we could estimate the effects of parental unemployment on delayed enrollment.

Section 2 provides the theoretical background of this study in the following two aspects: socio-economic research and statistical methodology. We drew upon the literature on children's education to discuss potential mechanisms behind the effects of parental job loss on delayed college enrollment. The necessity of survival analysis models for interval-censored data to analyze the KLIPS data used in this study is also explained. Section 3 provides details on several methodologies for handling interval censoring in the Cox proportional hazard model and describes the multiple imputation approach, which is the focus of this paper. Section 4 presents the simulation study for comparing the performance of the methodology for processing interval censoring and the results. In Section 5, the factors affecting children's enrollment in university are analyzed using data from the Korea Labor Panel Survey for the preceding 20 years, and their significance is examined. Section 6 discusses the implications of this study and suggestions for public policy in South Korea.

## 2. Theoretical Background

### 2.1. A Relationship between Parental Unemployment and College Entrance

Convincing evidence indicates that education is one of the key channels through which parental socio-economic status is transmitted to their children (Breen and Goldthorpe 1997; Kopycka 2021; Van de Werfhorst et al. 2003). Job loss threatens socio-economic status by negatively affecting future income, job security, and health (Baum 2003). Parental job loss is devastating in that it has intergenerational effects. In particular, children's educational achievement is susceptible to parental job loss. Involuntary parental unemployment arouses tension and the possibility of family disintegration (Charles and Stephens 2004; Jahoda et al. 2017), which leads to adverse consequences on children's academic achievements (Johnson et al. 2012). Becker (2009) showed that the likelihood of wealth transfer increases as the educational expenditure increases. This suggests that parental socio-economic background affects children's future income, which is mediated by children's educational achievements. Since long-term household income tends to decline after job loss regardless of re-employment, (Brand 2015; DiPrete and McManus 2000; Gangl 2006), educational expenditure can be limited for a certain period. Coelli (2011) and Kalil and Wightman (2011) found that decreased income because of the unemployment of parents affects their children's educational outcomes.

Many studies have revealed the adverse effects of parental job loss on children's college enrollment (Coelli 2011; Lindemann and Gangl 2019; Pan and Ost 2014). According to Lindemann and Gangl (2019), fathers' unemployment had a negative effect on their children's college admissions in Germany, but to a lesser extent than in the United Kingdom and United States. In contrast to countries such as the United States and United Kingdom, Germany has robust social protection measures in place against unemployment, and most colleges offer tuition assistance, thereby leading to the economic impact being minimal.

In South Korea, many studies have focused on the relationships between parental socio-economic status and children's education (Kim 2007; Shin 2010; Yang 2016). Kim (2007) found that the socio-economic status of the family affects the children's educational achievement, which is mediated by the parent–child relationship. Yang (2016) discovered that the phenomenon of educational pedigree, in which the academic achievement of students rises as the socio-economic status of their parents increases, was observed in all OECD member countries. The strength of the phenomenon varied according to the type of welfare in each country. In particular, the intergenerational transmission of parental socio-economic status was more prevalent in the liberal welfare system to which Korean education belongs. Shin (2010) divided socio-economic status into two groups using the case study method: high-education-middle class and low-education-working class. Then, they investigated the impact of parents' socio-economic backgrounds on their children's education in depth. While well-educated middle-class parents encouraged their children to study with the goal of gaining admissions into high-ranked colleges, less-educated working class parents tended to leave their children to make their own decisions in most cases. Ku (2003b) analyzed how family backgrounds, such as family structure, low income, and poverty, affect children's college admissions. The investigation found that having a single parent and living in poverty both had negative effects on children's college admissions; moreover, the situation of single parents was shown to have a more detrimental effect than that of poverty. Ku (2003a) analyzed the impact of parental unemployment on children's college admission. The study used three years (1998–2000) of data from KLIPS. Children who experienced parental unemployment at 16 to 18 years of age tended to have lower probabilities of obtaining college admission at 19 years of age.

As a measure to evaluate a child's educational achievement, most of the studies utilized whether or not to enroll in a university. Along with its rapid economic growth, the tertiary school enrollment rate of South Korea has increased sharply. In 1995, the global ranking of the tertiary school enrollment rate of South Korea was 9, which was indicative of huge growth from the ranking of 44 in 1980. The university entrance rate in Korea in 2019 was 69.8% higher than the OECD average of 44.9% (OECD 2020). Children who have experienced parental job loss may not be able to enter college immediately after graduating high school because of poor academic performance or financial issues, resulting from parental job loss. Since most Korean high school graduates enroll in college, children may be under pressure to enter college even later, when they resolve the relevant issues. This implies that parental job loss would affect not only the decision to enroll in college but also the timing of college enrollment. Therefore, we analyzed how parental job loss affects the timing of college enrollment. This analysis can help the government to develop and implement its education strategy on time by clearly understanding how parental job loss affects children's education.

*2.2. Characteristics of Time to Admission as Survival Data*

Survival analysis considers the time elapsing until the interest event occurs as a response variable. The time it takes to enroll in college can be seen as the survival time. In specific, the panel data of the Korea Labor Institute, which is collected annually from a survey administered to the same target audience, is appropriate for survival analysis methodology in that the time to enter university can be utilized as a response variable. In the panel data, right censoring occurs when no further investigation can be made because of the moving of, immigration of, or loss of contact with the household to be investigated.

The Cox proportional hazards regression model is the one of survival models to analyze the relationship between the event time and explanatory variables by dealing with the right censoring (Cox 1972). It is a semiparametric model because it assumes only the model for the regression coefficient without any assumptions on a baseline hazard function.

Other than being right-censored, the KLIPS data are interval-censored. Survival data are called interval-censored if a subject's survival time is only known within a certain specified time interval instead of being observed exactly. In the KLIPS data, the households' responses are annually recorded. If a subject enters university during a year when the subject's response is missing, the exact timing of the college entrance will be unknown and interval censoring will occur.

Since the interval-censored survival time can be considered as missing data, imputation methods can resolve the missing data issue. There are two types of imputation methods: single imputation and multiple imputation. While the single imputation method replaces the missing value as a single estimated value, the multiple imputation method pools multiple analysis results based on the multiple imputed data sets. Pan (2000) proposed the use of multiple imputation (Rubin 1987) for interval-censored data and the employment of Cox regression for the imputed data. As a nonparametric method, Turnbull (1976) suggested nonparametric survival functional estimation methods satisfying self-consistency. Groeneboom and Wellner (1992) proposed iterative convex minorant (ICM) algorithms to improve convergence speed of Turnbull's method. Wellner and Zhan (1997) combined an EM algorithm and an ICM algorithm as an EM-ICM algorithm. Finkelstein (1986) applied a Newton–Raphson algorithm to the Cox regression by adding covariates to the model in a nonparametric way. Zeng et al. (2016) formulated a semiparametric survival model called IntCens for interval-censored survival time with time-dependent covariates. They employed nonparametric maximum likelihood estimation with an EM-type algorithm.

## 3. Data and Methodology

### 3.1. Cox Proportional Hazards Regression

Survival models explore a relationship between a hazard function and a set of covariates. The Cox proportional hazards model assumes that the effect of a unit increase in a covariate is multiplicative to the hazard rate with a proportional hazard assumption. The proportional hazard model has a nonparametric form in that it does not assume any distribution for the survival time or specification for the baseline hazard function. Furthermore, because it assumes only the model for the regression coefficient $\beta_k$ and uses a parametric method to conduct estimations $\beta_k$, it is a semiparametric model.

We will assume that $T$ is a non-negative continuous random variable representing the survival time or the time to a specific event (e.g., time to admission). The hazard function at time $t$ is defined as

$$h(t) = \lim_{dt \to 0} \frac{P\{t \leq T \leq t + dt | T \geq t\}}{dt}.$$

The Cox regression specifies the hazard function of the $i$-th subject with covariates $x_i$ in the following manner.

$$h_i(t) = h(t|x_i) = h_0(t) \exp\left(\sum_{k=1}^{p} \beta_k x_{ik}\right)$$

The covariate vector is given by $x_i = (x_{i1}, x_{i2}, \cdots, x_{ip})$, and the baseline hazard function at time $t$ is denoted by $h_0(t)$. The regression coefficients $\beta = (\beta_1, \beta_2, \cdots, \beta_p)$ are estimated by maximizing the partial likelihood as $\hat{\beta}$, where the partial likelihood is given by

$$PL(\beta) = \prod_{i=1}^{n} \left[\frac{\exp(\beta' x_l)}{\sum_{l \in R(t_i)} \exp(\beta' x_l)}\right]^{\delta_i}. \tag{1}$$

$\delta_i$ is an indicator variable for the censoring of the $i$-th subject. $R(t_i)$ is a risk set that is exposed at any risk at time $t_i$, which includes subjects that have not experienced the event before $t_i$ and are not censored. A null hypothesis for $\beta$, $H_0 : \beta = \beta_0$, is tested using a Wald test, a likelihood ratio test, or a score test (Cox 1972) .

*3.2. Multiple Imputation for Interval-Censored Time*

The data with censoring are denoted by $D = \{(A_i, x_i), i = 1, \cdots, n\}$, where $A_i$ is $(L_i, R_i]$ and $x_i$ is a $p$-dimensional covariate vector. If the survival time $T_i$ is left-censored, $L_i = 0$; if the survival time is not censored, $L_i = R_i$. If it is right-censored, $R_i = \infty$. Since the partial likelihood function in Equation (1) can be calculated for the data with right censoring but not for the data with left or interval censoring, an additional step is necessary for the interval-censored cases in the KLIPS data.

We considered three approaches for dealing with the interval-censored cases: omitting the interval-censored cases, midpoint imputation, and multiple imputation. Midpoint imputation refers to imputing the interval-censored time to the event by using the midpoint of the interval $(L_i, R_i)$ as $(L_i + R_i)/2$. While midpoint imputation can be classified through simple imputation, multiple imputation is one of the probability-based imputation methods. Pan (2000) proposed the use of multiple imputation (Rubin 1987) for interval-censored data and the employment of Cox regression for imputed data.

For this paper, we used the MIICD package (Delord and Génin 2015) in the R program to implement the multiple imputation method. For the imputation of the interval-censored data, we considered the use of poor man's data augmentation (PMDA) or asymptotic normal data augmentation (ANDA). When there are few missing values, the PMDA methodology underestimates the actual variability (Wei and Tanner 1991), and ANDA is recommended for the imputation algorithm. In addition, when the number of truncated data is small, the regression coefficient converges to 0, so it is recommended to use "Link estimate" instead of the Breslow method to estimate the baseline survival function (Pan 2000).

This method uses an iterative algorithm and generates multiple imputed data sets. The subscript $(k)$ and the superscript $(i)$ represent the $k$-th imputed data set and the $i$-th iteration, respectively. Let us say $(T_{(k)}, \delta_{(k)}, X)$ represent $m$ sets of possibly right-censored values for $k = 1, \ldots, m$. $T$ is the observed event time, $\delta$ is whether or not it is censored, and $X$ is the set of covariates. The multiple imputation method proposed by Pan converts interval-censored data to right-censored data using the PMDA or ANDA method and then calculates it through the partial likelihood ratio. The detailed algorithm is as follows. Without loss of generality, only one explanatory variable $x_j$ and the corresponding regression coefficient $\beta$ are considered.

1. In the $i$-th iteration, the estimates for the regression coefficient and the baseline survival function are denoted by $\hat{\beta}^{(i)}$ and $\hat{S}_0^{(i)}$. Note that the starting value is $\hat{\beta}^{(0)} = 0$. After assuming a uniform distribution for $L_j$ and $R_j$ in the $m$ sets, the failure time $X_j$ is randomly generated and designated as an imputed value. This is expressed as $T_{(k),j} = X_j$ and $\delta_{(k),j} = 1$. The baseline survival probability $\hat{S}_{0,(k)}^{(0)}$ is the Breslow estimate of the baseline survival probability for the $k$-th replaced data set.

2. We generate $m$ sets of imputed data $\{T_{(1)}, \delta_{(1)}, X\}, \cdots, \{T_{(m)}, \delta_{(m)}, X\}$ which are possibly right-censored as follows. For each observation $L_j$, $R_j$, and $x_j$, $j = 1, \cdots, n, m$ sets created as right-censored data by replacing interval-censored time is empirically, and $\hat{S}_0$ in the second step is discrete assumed as follows:
   Each object has $((L_j, R_j, X_j)), j = 1, \ldots, n$ and $k = 1, \ldots, m$, if $V_j < \infty$. Samples $Y_j$ are from the $[\hat{S}_0^{(i)}]^{\exp(Z_j \hat{\beta}^{(i)})}$ distribution, under the condition that $\{L_j < T_j < R_j\}$, let $\{L_j < T_j < R_j\}$, $T_{(k),j} = Y_j$ and $\delta_{(k),j} = 1$. In the case of $R_j = \infty$, $T_{(k),j} = L_j$ and $\delta_{(k),j} = 0$.

   $(L_j, R_j)$ is interval-censored time and the $i$-th base survival function $[\hat{S}_0^{(i)}]^{\exp(Z_j \hat{\beta}^{(i)})}$ is $\{p_1, \ldots, p_{k_j}\}$ following the probability mass function at $\{t_1, \ldots, t_{k_j}\}$. Here, the failure

time $Y_j$ is randomly proportional to the probability at $\{t_1, \ldots, t_{k_j}\}$ with the probability mass function value $\{p_1, \ldots, p_{k_j}\}$.

3.　Since all the interval-censored values are imputed, the Cox proportional hazard model can be employed. Through this, the regression coefficient estimate can be considered as being $\hat{\beta}_{(k)}^{(i)}$ and the covariance estimate can be considered as being $\widehat{\Sigma}_{(k)}^{(i)}$.

4.　$\left(T_{(k)}, \delta_{(k)}, X\right)$ denotes the $k$-th right-censored data of $m$ sets obtained through the imputation of the interval-censored data. $\hat{\beta}_{(k)}^{(i)}$ denotes the regression coefficients obtained by fitting a Cox proportional hazard model. The Breslow estimate $\hat{S}_{0,(k)}^{(i)}$ for the basis survival function is calculated based on $\left(T_{(k)}, \delta_{(k)}, X\right)$ and $\hat{\beta}_{(k)}^{(i)}$.

5.　In the $i$-th iteration, the $\hat{\beta}_{(k)}^{(i)}$ of $m$ sets is summed and divided by $m$, which is denoted by $\beta^{(i+1)}$. In this way, the basis survival function is also obtained. The covariance is the sum of the intragroup and intergroup imputation variances. This can be expressed as an equation as follows.

$$\hat{\beta}^{(i+1)} = \frac{1}{m} \sum_{k=1}^{m} \hat{\beta}_{(k)}^{(i)}, \quad \hat{S}_0^{(i+1)} = \frac{1}{m} \sum_{k=1}^{m} \hat{S}_{0,(k)}^{(i)}$$

$$\widehat{\Sigma}^{(i+1)} = \frac{1}{m} \sum_{k=1}^{m} \widehat{\Sigma}_{(k)}^{(i)} + \left(1 + \frac{1}{m}\right) \frac{\sum_{k=1}^{m} \left[\hat{\beta}_{(k)}^{(i)} - \hat{\beta}^{(i+1)}\right]^2}{m-1}$$

Finally, it repeats from the first until the $\hat{\beta}^{(i)}$ converges.

The ANDA method includes a variation in the fifth step of the PMDA above. In the fifth step, the normal distribution is assumed with a mean vector of the regression coefficients and a covariance matrix of the covariances are obtained from the $k$-th set; furthermore, the estimated values of the regression coefficients are obtained.

$$\hat{g}^{(i+1)}(\beta) = \frac{1}{m} \sum_{k=1}^{m} N\left(\hat{\beta}_{(k)}^{(i)}, \widehat{\Sigma}_{(k)}^{(i)}\right)$$

## 4. Simulation Study

In order to find an appropriate imputation method for the KLIPS data with interval censoring, the imputation performance of the imputation methods for interval-censored data, the simple omission of the interval-censored data, the midpoint imputation, and the multiple imputation were compared. We compared the models in terms of model estimation, which is measured by the mean squared error (MSE). The mean squared error (MSE) was calculated to evaluate the performance of the three imputation methods. The MSE is a method for measuring the accuracy of the estimated regression coefficient value.

$$MSE = \frac{1}{p} \sum_{j=1}^{p} (\hat{\beta}_j - \beta_j)^2$$

We employed the IntCens method (Zeng et al. 2016) to compare the estimation accuracy with other methods. For both simulation and real data sets, the IntCens method failed because of a singularity issue. Since the parental unemployment rate was only 2.63%, the simulated data contain only a few cases having unemployment experience. This may cause a numerical singularity problem. To resolve the singularity issue, we added random noise on the simulated response variable values. The simulation results are presented in the Appendix A. The MSE values of the IntCens were the largest regardless of simulation settings. On the top of that, its Monte Carlo standard deviation of the MSE values was the

largest, which implies unstable model estimation. We concluded that the IntCens is not suitable for the KLIPS data. More results can be found in Appendix A.

### 4.1. Simulation Settings

In order to generate the simulation data, we mimicked the censoring rate of the KLIPS data, while we considered several scenarios for the interval censoring rate from the low values to the high values. The right censoring rate was fixed at 20%, and the interval censoring rate was fixed as 15%, 30%, or 45%. The simulation data were sampled with replacement from the KLIPS data of 376 subjects whose parental education levels reached college or higher. We considered samples sizes of 100, 300, and 1000 to compare the imputation methods for different cases.

The baseline hazard function was generated from the exponential distribution and the Weibull distribution. A nonparametric method was also employed to mimic a case where the data were generated from Cox regression; this is known as the flexible-hazard method. For the noncensored cases, a uniform distribution was used to generate the left and right bound times $L$ and $R$. If the left and right bound times were not the same, the case was regarded as interval-censored time. The survival time was generated assuming that the true regression coefficient was $\beta = (0.058, 0.0446, -0.8758, -0.052)$, which can be obtained by repeating 100 times for the MIICD package to which the multiple imputation method was applied.

### 4.2. Simulation Results

The MSE values are summarized in Table 1, where the exponential distribution is assumed. When the baseline hazard function follows an exponential distribution, the MSE tends to decrease as the sample size increases, regardless of the interval censoring rate. Furthermore, for a given sample size and interval censoring rate, the MSE of the multiple imputation method is slightly lower than that of the midpoint imputation and omission method for the sample size of 100, 300. In the cases of the sample sizes of 1000, there is no difference between the MSE of the multiple imputation and the midpoint imputation, and the MSE of the midpoint imputation and MSE of the multiple imputation are lower than those of the omission.

**Table 1.** MSE values are presented when the exponential distribution is assumed.

| right censoring | 20% | | | | | | | | |
|---|---|---|---|---|---|---|---|---|---|
| sample size | 100 | | | 300 | | | 1000 | | |
| interval censoring | 15% | 30% | 45% | 15% | 30% | 45% | 15% | 30% | 45% |
| omission | 0.444 | 0.432 | 0.473 | 0.195 | 0.216 | 0.190 | 0.088 | 0.095 | 0.105 |
| midpoint imputation | 0.337 | 0.309 | 0.317 | 0.184 | 0.191 | 0.157 | 0.080 | 0.080 | 0.080 |
| multiple imputation | 0.333 | 0.305 | 0.305 | 0.188 | 0.187 | 0.154 | 0.080 | 0.080 | 0.080 |

The simulation results are summarized in Table 2, where the right censoring rate is set as 20% and the Weibull distribution is assumed for the baseline hazard function. As in the case of the exponential distribution, the MSE value tends to decrease as the sample increases, regardless of the interval censoring rate. Furthermore, for a given sample size and interval censoring rate, the MSE of the multiple imputation is lower than that of the midpoint imputation when the sample size is 100 or 300. The MSE of the midpoint imputation is lower than that of the multiple imputation and omission for some cases. For a sample size of 1000, there is little difference between the MSE of the multiple imputation and that of the midpoint imputation, and in some cases, the MSE of the midpoint imputation is lower than that of the multiple imputation.

The simulation results are summarized in Table 3, where the right censoring rate is fixed at 20% and the flexible-hazard method is assumed for the baseline hazard function.

As in the case of assuming the flexible-hazard method, the MSE value tends to decrease as the sample is enlarged, regardless of the interval censoring rate. Furthermore, when the sample size and the interval censoring rates are the same, the MSE of the multiple imputation is lower than that of the midpoint imputation if the sample size is 100, 300, or 1000.

**Table 2.** MSE when the Weibull distribution is assumed.

| right censoring | | | | 20% | | | | | |
|---|---|---|---|---|---|---|---|---|---|
| sample size | | 100 | | | 300 | | | 1000 | |
| interval censoring | 15% | 30% | 45% | 15% | 30% | 45% | 15% | 30% | 45% |
| omission | 0.323 | 0.341 | 0.359 | 0.223 | 0.234 | 0.239 | 0.075 | 0.081 | 0.090 |
| midpoint imputation | 0.293 | 0.283 | 0.278 | 0.221 | 0.225 | 0.218 | 0.071 | 0.071 | 0.070 |
| multiple imputation | 0.295 | 0.283 | 0.276 | 0.218 | 0.222 | 0.216 | 0.071 | 0.071 | 0.071 |

**Table 3.** MSE when the flexible-hazard method is assumed.

| right censoring | | | | 20% | | | | | |
|---|---|---|---|---|---|---|---|---|---|
| sample size | | 100 | | | 300 | | | 1000 | |
| interval censoring | 15% | 30% | 45% | 15% | 30% | 45% | 15% | 30% | 45% |
| omission | 0.740 | 0.775 | 0.952 | 0.249 | 0.313 | 0.406 | 0.061 | 0.075 | 0.090 |
| median imputation | 0.694 | 0.710 | 0.788 | 0.233 | 0.212 | 0.220 | 0.056 | 0.055 | 0.057 |
| multiple imputation | 0.689 | 0.683 | 0.726 | 0.231 | 0.212 | 0.216 | 0.056 | 0.055 | 0.055 |

The multiple imputation method showed a lower MSE than the midpoint imputation and omission when the sample size was 100, and the right censoring rate was fixed at 20%. However, the sample sizes of 300 and 1000 showed similar MSEs to those of the midpoint imputation, regardless of the interval censoring rate. The midpoint imputation is affected by the sample size, so the MSE decreases as the sample gets larger; however, the multiple imputation method shows a low residual regardless of the sample size, so it is a good method for imputing interval censoring with a robust model. Therefore, we proceed to use the Cox proportional hazards model with a multiple imputation method, which shows a better model estimation accuracy.

## 5. Data Analysis

### 5.1. Data

This study's utilized data were taken from the Korean Labor & Income Panel Study (KLIPS), and the survey was administered to households living in urban areas and their household members. The members of the panel sample were household members from a sample of 5000 households. The KLIPS, a longitudinal survey, followed up on the subjects once a year to gain data about economic activities, labor market movement, income activities and consumption, education and vocational training, and social life.

The data used for the analysis were based on 20 years of panel data, which were collected from surveys conducted between 1998 and 2017 as part of the KLIPS. We analyzed the effect of parental job loss on children's college entrance. Since the Korean Labor Panel Survey is conducted in urban areas, the survey data were limited to apply analysis results to rural areas. Of the 989 subjects, 58 (5.9%) were right-censored and 79 (7.9%) were interval-censored time.

Among the variables of KLIPS, the householder's education level, gender of the first child, gender of the householder, poverty, employment status of the first child's parents,

double income status of the first child's parents, and the number of household members were selected as covariates based on the study by Ku (2003a). The descriptive statistics of the chosen variables are summarized in Table 4. However, the Fisher exact test results showed that the correlation between household poverty and parental unemployment experience was significantly high (*p*-value = 0.025); furthermore, the poverty variable was excluded from the real data analysis. Assuming that the effect of parental unemployment may vary depending on the household head's academic background, we divided the sample into two subsets: a sample where the household head's education level included the achievement of a high school diploma or a lower qualification and a sample where the household head's education level included the achievement of a college degree or a higher qualification.

**Table 4.** Variables and their descriptive statistics.

| | | **Frequency** | **Proportion (%)** |
|---|---|---|---|
| | middle school graduation or less (1) | 153 | 15.5% |
| education level of householder | high school graduation or less (2) | 458 | 46.3% |
| | college graduation or more (3) | 378 | 38.2% |
| sex of the first child | male (0) | 500 | 50.6% |
| | female (1) | 489 | 49.4% |
| poverty | no (0) | 926 | 93.6% |
| | yes (1) | 63 | 6.4% |
| whether parents are unemployed | no (0) | 963 | 97.4% |
| | yes (1) | 26 | 2.6% |
| double income | no (0) | 109 | 11.0% |
| | yes (1) | 880 | 89.0% |
| | 2 | 19 | 1.9% |
| | 3 | 130 | 13.1% |
| the number of household members | 4 | 635 | 64.2% |
| | 5 | 173 | 17.5% |
| | 6 | 32 | 3.2% |
| | right censoring | 58 | 5.9% |
| censoring | interval censoring | 79 | 7.9% |
| | no censoring | 852 | 86.2% |
| Total | | 989 | 100% |

### 5.2. Analysis Results

In Table 5, the model's estimation results were summarized for the sample where household heads' education levels included qualifications under or equal to the achievement of a high school diploma. The first child's probability of being admitted to a college under circumstances of parental unemployment was 58.4% lower than those who don't experience parental unemployment in the multiple imputation. The variables of parental unemployment were significant in the cases of omission and multiple imputation at a significance level of 5%. Unlike other imputation way, the midpoint imputation produced a positive coefficient regarding the direction of the effect of the parental unemployment variable. Table 6 shows the model estimation results for the sample where household heads' education levels included qualifications that were college graduation or more. The first child's probability of being admitted to a college under circumstances of parental unemployment was 57.5% lower than that in the other cases. The variables of parental unemployment were significant in all the imputation methods at a significance level of 5%. When the interval-censored data were omitted, in the case of double-income households,

the probability of being admitted to a college was 46% higher than that in the other cases. This shows that use of an inappropriate imputation method for interval censoring (such as simple omission or midpoint imputation) could distort the data analysis results.

**Table 5.** Comparison of model estimation for the sample whose householder graduated high school or less.

| | High School Graduation or Less | | | | | | | | | | | |
| | Omission | | | | Midpoint Imputation | | | | Multiple Imputation | | | |
| | n = 576; Number of Events = 524 | | | | n = 610; Number of Events = 558 | | | | n = 610; Number of Events = 558 | | | |
| | $\hat{\beta}$ | exp ($\hat{\beta}$) | se($\hat{\beta}$) | *p*-Value | $\hat{\beta}$ | exp ($\hat{\beta}$) | $\hat{\beta}$) | *p*-Value | se($\hat{\beta}$) | exp ($\hat{\beta}$) | se($\hat{\beta}$) | *p*-Value |
|---|---|---|---|---|---|---|---|---|---|---|---|---|
| sex | 0.108 | 1.114 | 0.108 | 0.316 | 0.074 | 1.077 | 0.086 | 0.349 | 0.055 | 1.056 | 0.113 | 0.628 |
| double income | 0.285 | 1.330 | 0.186 | 0.125 | 0.138 | 1.148 | 0.148 | 0.370 | 0.068 | 1.071 | 0.195 | 0.727 |
| whether parents are unemployed | −0.787 | 0.455 | 0.384 | 0.040 ** | 0.186 | 1.204 | 0.254 | 0.330 | −0.877 | 0.416 | 0.391 | 0.025 ** |
| the number of household members | −0.091 | 0.913 | 0.085 | 0.283 | −0.043 | 0.958 | 0.063 | 0.409 | −0.059 | 0.943 | 0.088 | 0.502 |

** denotes significance at 5% level.

**Table 6.** Comparison of significance of regression coefficients for the households whose householder graduated college or more.

| | College Graduation or More | | | | | | | | | | | |
| | Omission | | | | Midpoint Imputation | | | | Multiple Imputation | | | |
| | n = 333; Number of Events = 315 | | | | n = 376; Number of Events = 357 | | | | n = 376; Number of Events = 357 | | | |
| | $\hat{\beta}$ | exp ($\hat{\beta}$) | se($\hat{\beta}$) | *p*-Value | $\hat{\beta}$ | exp ($\hat{\beta}$) | se($\hat{\beta}$) | *p*-Value | $\hat{\beta}$ | exp ($\hat{\beta}$) | se($\hat{\beta}$) | *p*-Value |
|---|---|---|---|---|---|---|---|---|---|---|---|---|
| sex | 0.114 | 1.120 | 0.108 | 0.292 | 0.108 | 1.114 | 0.108 | 0.316 | 0.070 | 1.073 | 0.112 | 0.530 |
| double income | 0.381 | 1.464 | 0.188 | 0.043 ** | 0.285 | 1.330 | 0.186 | 0.125 | 0.031 | 1.032 | 0.213 | 0.883 |
| whether parents are unemployed | −0.753 | 0.471 | 0.383 | 0.049 ** | −0.787 | 0.455 | 0.384 | 0.040 ** | −0.857 | 0.425 | 0.388 | 0.027 ** |
| the number of household members | −0.119 | 0.888 | 0.888 | 0.176 | −0.091 | 0.913 | 0.085 | 0.283 | −0.040 | 0.961 | 0.088 | 0.649 |

** denotes significance at 5% level.

*5.3. Comparison of Predictive Performance According to the Imputation Method*

We used the time-dependent receiver operating characteristic (ROC) curve to evaluate the predictive power of survival data instead of a simple ROC curve, which is used for evaluating the predictive power of binomial response variables. The area AUC ($t$) under the time-dependent ROC curve can be calculated at each time point $t$. The integrated AUC (iAUC) was used to compare the prediction performance of statistical methods for interval censoring.

Figures 1 and 2 show ROC curves at a certain time point $t$ when interval-censored data are imputed by the multiple imputation method. Figure 1 is for a child whose household head has the achievement of 328 a high school diploma or lower qualification. Figure 2 is for a child whose household head has achieved a college degree or a higher qualification. When $t$ is less than 5, the ROC curves of the two samples show similar predictive performance. When $t$ is greater than or equal to 5, the ROC curve of a sample with a higher education level shows better predictive performance.

The closer iAUC is to 1, the better the model is; the closer it is to 0.5, the less accurate the model is. The iAUC of the sample where household heads had educational qualifications that were below or equal to the achievement of a high school diploma was estimated as 0.51 based on five-fold cross-validation. The iAUC of the sample where household heads had educational qualifications that were equal to or higher than the achievement of a college degree was estimated as 0.53 based on the five-fold cross-validation.

This result implies that the predictive performance of the Cox regression model is very poor. Thus, this estimated model is limited to only interpretation, not prediction.

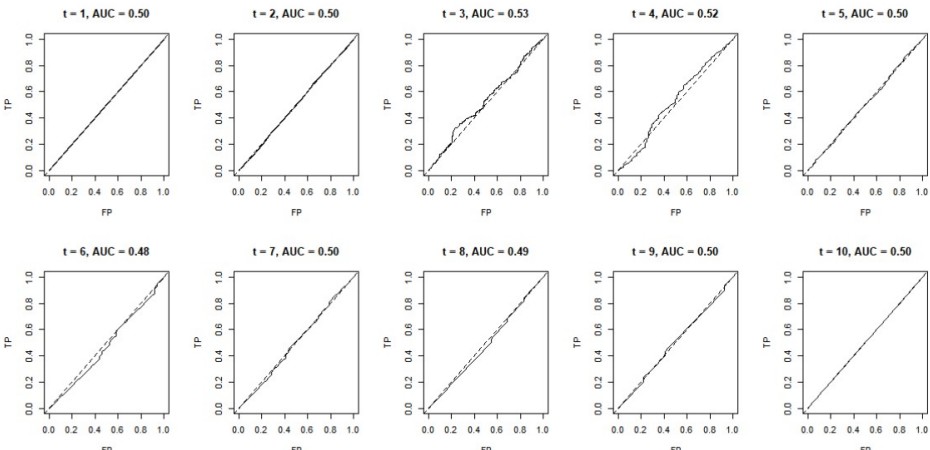

**Figure 1.** ROC curves at a certain time point *t* when interval-censored data are imputed by the multiple imputation method.

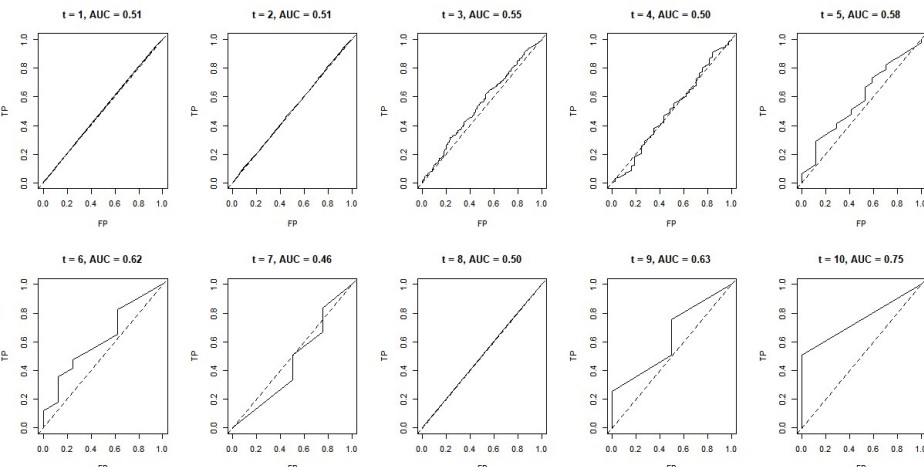

**Figure 2.** ROC curves at a certain time point *t* when interval-censored data are imputed by the multiple imputation method.

## 6. Conclusions

Using 20 years' worth of data (1998 to 2017) from the Korean Labor Panel Data (KLIPS), we analyzed how college admissions could be affected by parental unemployment status when the first child in a household was aged 18 years and preparing for college. Suppose the child was admitted to the college in 2018, and the household answered the survey in 2020, except for the information about the child's college admission year. In this case, the panel data are interval-censored data since the researcher was unaware of the exact time of admission because of a lack of response. We considered three imputation methods for interval censoring: simple omission, the midpoint imputation, and the multiple imputation proposed by Pan. In order to choose an appropriate imputation method for this data, we ran extensive simulation studies. Mean squared errors (MSEs) were compared to evaluate the performance of the imputation methods. For the simulation study, 100, 300, and 1000 samples were resampled with replacement from the real data of 376 subjects whose parental education qualifications included college graduation or a higher qualification. The right censoring rate was set as 0.2, and the interval censoring rate varied from 0.15 to 0.45. Overall, the model estimation accuracy of the multiple imputation method was found to be higher than that of other imputation methods. The estimation accuracy midpoint imputation was affected by the sample size, so the MSE decreased as the sample grew larger. On the other hand, the multiple imputation method showed a low residual regardless of the sample size.

Therefore, we can conclude that multiple imputation is a good method for ensuring robust model estimation.

The real data analysis showed that the effect of the variable "whether or not the parents are unemployed" on the time taken to be admitted to a college was significant only when the householder's academic background was higher than and equal to college graduation. When the interval-censored data were removed, the "double income" and "parents' unemployment" variables became significant. The first children of double-income parental households had a 46% higher probability of entering a college than others. On the contrary, when the first children experienced their parents' unemployment at the age of 18, the probability of college admission was reduced nearly by 53% compared to cases where they did not. Therefore, our study suggests that college entrance is affected by parental financial status—in particular for households where the householder's academic background is higher than and equal to college graduation. Children having well-educated parents tend to have strong will to get into good universities (Shin 2010). After experiencing parental job loss, the children cannot get admission to the colleges they have aimed for, leading to delayed college enrollment.

In the past, the economic crisis contributed to parental unemployment. These days, government policies to prevent infectious diseases such as COVID-19 can contribute to parental unemployment. Parental job loss may happen more frequently because of the long-term economic recession and recurring epidemics. It is crucial to understand the effects of parental unemployment on children's academic achievements so that the government can develop a policy targeting adolescents who cannot expect parental help. The government should develop a policy in the education system offering psychosocial support. First, the government should improve the quality of education. Many South Korean students study at availing profit-making private institutes instead of regular school programs (Yoo 2021). If regular school programs play their standard roles, the financial contraction resulting from parental job loss would not affect the children's academic performance as much. Additionally, the government and municipality need to provide psychosocial support to the households experiencing job loss. As per the OECD database, the proportion of dual-income households in South Korea having children less than 14 years was 29.4%, half the average of that of OECD countries, being 60.7% in 2018. Since most households in South Korea rely on the earnings of the head of the household, parental unemployment greatly impacts a family's economic situation Lim (2020). Therefore, children experiencing parental job loss will suffer from the traumatic effects of job loss (Wightman 2012).

**Author Contributions:** Conceptualization, H.K. and E.L.; methodology, H.K. and E.L.; software, H.K. and S.K.; formal analysis, H.K. and E.L.; data curation, H.K.; writing—original draft preparation, H.K., E.L. and S.K.; writing—review and editing, H.K., E.L. and S.K.; supervision, E.L. All authors have read and agreed to the published version of the manuscript.

**Funding:** This study was funded by the 2019 research fund of Chungnam National University.

**Institutional Review Board Statement:** Not applicable

**Informed Consent Statement:** Informed consent was obtained from all subjects involved in the study.

**Data Availability Statement:** The data are available in the following website: https://www.kli.re.kr/klips_eng/index.do (accessed on 1 September 2019).

**Conflicts of Interest:** The authors declare no conflict of interest.

## Appendix A

We generated simulation data in the same manner in Section 4.1. Since the IntCens method had numerical singularity errors, we added random noise on the right bound time $R$ of the simulated response variable values. The random noise was generated from the normal distribution with a mean of 0 and a standard deviation of 1. The MSE values of the IntCens were the largest regardless of simulation settings. On top of that, its Monte

Carlo standard deviation of the MSE values was the largest, which implies unstable model estimation. We concluded that the IntCens was not suitable for the KLIPS data.

**Table A1.** MSE values are presented when the exponential distribution is assumed.

| right censoring | 20% | | | | | | | | |
|---|---|---|---|---|---|---|---|---|---|
| sample size | | 100 | | | 300 | | | 1000 | |
| interval censoring | 15% | 30% | 45% | 15% | 30% | 45% | 15% | 30% | 45% |
| omission | 0.905 | 0.495 | 0.552 | 0.207 | 0.141 | 0.149 | 0.061 | 0.058 | 0.072 |
| midpoint imputation | 0.857 | 0.397 | 0.448 | 0.192 | 0.137 | 0.133 | 0.059 | 0.053 | 0.062 |
| multiple imputation | 0.775 | 0.392 | 0.441 | 0.200 | 0.144 | 0.140 | 0.064 | 0.060 | 0.065 |
| IntCens | 2.887 | 0.723 | 1.144 | 0.485 | 0.292 | 0.363 | 0.174 | 0.189 | 0.167 |

**Table A2.** MSE when the Weibull distribution is assumed.

| right censoring | 20% | | | | | | | | |
|---|---|---|---|---|---|---|---|---|---|
| sample size | | 100 | | | 300 | | | 1000 | |
| interval censoring | 15% | 30% | 45% | 15% | 30% | 45% | 15% | 30% | 45% |
| omission | 0.619 | 0.327 | 0.980 | 0.127 | 0.198 | 0.221 | 0.049 | 0.054 | 0.067 |
| midpoint imputation | 0.468 | 0.314 | 0.686 | 0.122 | 0.163 | 0.188 | 0.045 | 0.045 | 0.055 |
| multiple imputation | 0.486 | 0.344 | 0.688 | 0.116 | 0.166 | 0.187 | 0.049 | 0.048 | 0.060 |
| IntCens | 0.818 | 0.946 | 1.495 | 0.330 | 0.444 | 0.434 | 0.155 | 0.169 | 0.167 |

**Table A3.** MSE when the flexible-hazard method is assumed.

| right censoring | 20% | | | | | | | | |
|---|---|---|---|---|---|---|---|---|---|
| sample size | | 100 | | | 300 | | | 1000 | |
| interval censoring | 15% | 30% | 45% | 15% | 30% | 45% | 15% | 30% | 45% |
| omission | 0.502 | 0.708 | 1.172 | 0.197 | 0.250 | 0.217 | 0.040 | 0.054 | 0.071 |
| midpoint imputation | 0.484 | 0.530 | 0.643 | 0.163 | 0.128 | 0.150 | 0.035 | 0.038 | 0.051 |
| multiple imputation | 0.490 | 0.485 | 0.605 | 0.163 | 0.109 | 0.133 | 0.036 | 0.034 | 0.051 |
| IntCens | 1.156 | 1.092 | 1.481 | 0.470 | 0.456 | 0.555 | 0.255 | 0.203 | 0.177 |

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
