# Peer review of "Cox Proportional Hazards Regression for Interval-Censored Data with an Application to College Entrance and Parental Job Loss"

_economies, doi:10.3390/economies10090218_

Round 1

Reviewer 1 Report

The paper presents an analysis of the effect of parental job loss on children’s admission to a college and university.

The paper contains adequate literature review on the investigated topic, presented especially in the Introduction, but the Theoretical background should be improved with new papers studied. This will contribute to the improvements of the final references.’

I recommend moving a part of the information presented in the Introduction in Theoretical background section and complete this section with new research studies. In Introduction, the authors should present the importance of the study, the relevance to the readers, the novelty of the paper.

The methodology is very well described and is appropriate considering the paper aim. The Section 3 should be named: Methodology. In the Section 3.1 the authors should include citations.

The authors should include in the study an interpretation of the data presented in table 4.

The results are clearly presented for the aim established.

Policy implications research, practice and society are not presented. Some recommendations to regulators should be added in order to explain how they can benefit from the findings.

The paper is online here: https://assets.researchsquare.com/files/rs-1611050/v1_covered.pdf?c=1651514641 

Reviewer 2 Report

Please see attached note.

Round 2

Reviewer 1 Report

The authors have improved the paper according to reviewers recommendations. Congratulations for the paper.

Reviewer 2 Report

Accept